Histone H3K9 demethylase JMJD2B/KDM4B promotes osteogenic differentiation of bone marrow-derived mesenchymal stem cells by regulating H3K9me2 on RUNX2

Kang Pan 1
Wu Zhiming 2
Huang Yuxi 1
Luo Zhen 1
Huo Shaochuan 3
Chen Qunqun chenqunqun718@163.com 1 4 5
1 Guangzhou University of Chinese Medicine , Guangzhou , China
2 University Medical Center Utrecht , Utrecht , Netherlands
3 Shenzhen Hospital (Futian) of Guangzhou University of Chinese Medicine , Shenzhen , China
4 The Third Affiliated Hospital, Guangzhou University of Chinese Medicine , Guangzhou , China
5 Guangdong Research Institute for Orthopedics & Traumatology of Chinese Medicine , Guangzhou , China
Ozdag Sevgili Hilal
Electronic publication date: 2022 Oct 5
Publication date: 2022
Volume: 10
Electronic Location ID: e13862
Received 2021 Nov 29; Accepted 2022 Jul 18
Copyright: ©2022 Kang et al.
Copyright year: 2022
Copyright holder: Kang et al.
License: This is an open access article distributed under the terms of the Creative Commons Attribution License, which permits unrestricted use, distribution, reproduction and adaptation in any medium and for any purpose provided that it is properly attributed. For attribution, the original author(s), title, publication source (PeerJ) and either DOI or URL of the article must be cited.
License URL: https://creativecommons.org/licenses/by/4.0/

Keywords: Osteogenesis, JMJD2B, hBMSCs, Epigenetics, KDM4B

Funding: Natural Science Foundation of Guangdong Province 2018A030310606 This work was funded by the Natural Science Foundation of Guangdong Province (grant numer 2018A030310606 to Qunqun Chen). The funders had no role in study design, data collection and analysis, decision to publish, or preparation of the manuscript.

==============================
Background

A variety of proteins including epigenetic factors are involved in the differentiation of human bone marrow mesenchymal stem cells. These cells also exhibited an epigenetic plasticity that enabled them to trans-differentiate from adipocytes to osteoblasts (and vice versa) after commitment. Further in-depth study of their epigenetic alterations may make sense.

Methods

Chromatin Immunoprecipitation-PCR (ChIP-PCR) was used to detect the methylation enrichment status of H3K9me2 in the Runx2 promoter, alizarin red and alkaline phosphatase (ALP) staining were used to detect osteogenic differentiation and mineralization ability, western blot and quantitative RT-PCR were used to measure the differential expression of osteogenesis-related proteins and genes. Recombinant Lentivirus mediated gain-of-function and loss-of-function study. The scale of epigenetic modification was detected by laser confocal.

Results

Our results showed that compared with human bone marrow mesenchymal stem cells (hBMSCs) without osteogenic differentiation treatment, hBMSCs after osteogenic differentiation significantly promoted osteogenic differentiation and mRNA expression such as JMJD2B/KDM4B, osteogenesis-related genes like Runx2 and FAM210A in hBMSCs cells, suggesting that upregulation of JMJD2B/KDM4B is involved in the promoting effect of osteogenesis. After overexpression and silencing expression of JMJD2B, we found a completely opposite and significant difference in mRNA expression of osteogenesis-related genes and staining in hBMSCs. Overexpression of JMJD2B/KDM4B significantly promoted osteogenic differentiation, suggesting that JMJD2B/KDM4B could promote osteogenesis. In addition, ChIP-PCR showed that overexpression of JMJD2B/KDM4B significantly reversed the methylation enrichment status of H3K9me2 in Runx2 promoter. Furthermore, overexpression of JMJD2B/KDM4B significantly reverses the inhibitory effect of BIX01294 on H3K9me2, suggesting that JMJD2B/KDM4B regulates the osteogenic differentiation of hBMSCs by changing the methylation status of H3K9me2 at the Runx2 promoter.

Conclusions

Taken together, these results suggest that JMJD2B/ KDM4B may induce the osteogenic differentiation of hBMSCs by regulating the methylation level of H3K9me2 at the Runx2 promoter.

Introduction

The research on the regulation of human bone marrow mesenchymal stem cells(hBMSCs) towards bone formation has been going on for decades. These cells exist in the bone marrow cavity and can differentiate into a variety of cell phenotypes, including osteoblasts, chondroblasts, stromal cells that support hematopoiesis, and adipocytes. The mainstream control methods found in current research can be roughly divided into the following categories: (1) Cell-to-cell regulation. For example, the co-culture of hAMSCs/hBMSCs mixed at the optimal ratio of 3/1 showed significantly higher cell proliferation, antioxidant properties, osteogenic and angiogenic differentiation than hBMSCs or HUVECs alone (Zhang et al., 2018). (2) Hormone. Like insulin (Zhang et al., 2020), in addition, there are some xenogeneic hormones that also play a role in inducing differentiation to a certain extent (You & Xu, 2020). (3) Biomaterials. Through anodization and subsequent dip coating treatment, two highly ordered nano-pits with two different sizes were successfully constructed on the surface of 316LSS and then stimulated osteogenesis and angiogenesis (Ni et al., 2020). Shrestha et al. (2020) also found that both the composition and form of the hydrogel can determine the success of tissue formation, and these two factors have a complex interaction with cell behavior and matrix deposition. This has important implications for tissue engineering. In addition, a natural osteoinductive biomaterial nacre shell also has a similar function (Green, Kwon & Jung, 2015). (4) Large amounts of Chinese medicine monomers. e.g. Polydatin Chen et al., 2019a; Chen et al., 2019b; Shen et al., 2020), Chrysosplenetin (Hong et al., 2019), Hydroxysafflor yellow A (Wang et al., 2021) and Icariin (Xu et al., 2021). In addition, there are many kinds of exosome research. Such as, miR-375 (Chen et al., 2019a; Chen et al., 2019b), There are also some regulatory factors that are inconvenient to be placed elsewhere. Classics such as Ocn, Opn, Osx, and some newly discovered ones such as TIMP-1 (Liang et al., 2019), brain-derived neurotrophic factor (Liu et al., 2018). Some researchers have also discovered the epigenetic mechanism of hBMSCs (Cho et al., 2005; El-Serafi, Richard Oreffo & Roach, 2011; Lee et al., 2021). Some changes in the cellular environment can also cause hBMSCs to differentiate towards osteogenic direction. Pulsed electromagnetic field (PEMF) has also been successfully applied to accelerate fracture repair since 1979 (Fu et al., 2014). There are so many ways to regulate the directed differentiation of hBMSCs. Research on the effectiveness of promoting osteogenesis has become less attractive. It is also not easy to identify which type of regulatory means is most effective in promoting the osteogenic differentiation of hBMSCs. Therefore, we turned our attention to the changes in the cells themselves caused by various ways of promoting osteogenic differentiation. Most interestingly, we found that during the MSCs-transdifferentiation process were in relationship to the profiles of histone marks (Meyer et al., 2016). Changes of this type are called epigenetic modifications.

Epigenetic refers to genetic changes in gene expression without altering the DNA sequence, including DNA methylation, histone modification and small non-coding RNA related regulation (Wu & Sun, 2006), Lyu et al. (2019) found that large-grit, acid-etched (SLA) method-treated surfaces and mechanically processed surfaces have different effects on genome-wide DNA methylation and osteogenic pathway enrichment in hBMSCs. Among the existing epigenetic regulation of hBMSCs, MI192 and its regulated HDAC enzyme (Man et al., 2021b), Setd7 (Yin et al., 2018), CUDC-907 (Ali et al., 2017), RG108 (Assis et al., 2018), have been discovered. Man et al. (2021a) significantly changed the epigenetic function of osteoblasts by using the HDAC inhibitor trichostatin A (TSA) to reduce HDAC activity and increase histone acetylation. Accelerate their mineralization and promote the osteoinductive ability of secretory extracellular vesicles (EVs). Epigenetics is reversible and susceptible to factors such as environment and drugs. There are many traditional Chinese medicines that can promote the differentiation of hBMSCs towards the osteogenic direction. Wang et al. (2021) discovered that one of the essential compounds of safflower: Hydroxysafflor yellow A, could promote osteogenesis and bone development via epigenetically regulating β-catenin and prevent ovariectomy-induced bone loss. Some scholars have also discovered that the epigenetic landscape of 3D cell models of human primary articular chondrocytes (hPACs) and human bone-marrow derived mesenchymal stem cells (hBMSCs) exhibits a huge difference in DNA methylation landscape (Bomer et al., 2016). This may indicate the future research direction of tissue engineering. In our opinion, epigenetics has been fairly close to the osteogenic differentiation nature of hBMSCs. In another recent epigenetic study we found that vascular smooth muscle cells (VSMCs) exhibit osteoblast-like characteristics in response to various stimuli such as oxidized cholesterol and inflammation. In the study of Kurozumi et al. (2019), it was found that IL-6 and sIL-6R induces STAT3-dependent differentiation of human VSMCs into osteoblast-like cells through JMJD2B/KDM4B -mediated histone demethylation of Runx2. This had caused us to think about the osteogenic epigenetic regulation mechanism of hBMSCs based on Runx2. To further investigate the role of JMJD2B/KDM4B in regulating the differentiation of hBMSCs into osteoblasts, we performed the following experiments.

Materials & Methods

Cell culture

Human bone marrow-derived mesenchymal stem cell line was obtained from Saliai Biotechnology (G02007; Saliai Biotechnology, Guangzhou, China). Cells were maintained in α-MEM (Gibco, Waltham, MA, USA) supplemented with 10% fetal bovine serum (FBS), 50 µM ascorbate, 1% penicillin-streptomycin, and 10 mM β-glycerophosphate in a humidified atmosphere with 5% CO2 at 37 °C. Cells at the third or fourth passage were used in the experiments. For osteogenesis treatment, cells were cultured in human bone marrow mesenchymal stem cells osteogenic medium (HUXMA-90021; Cyagen).

Plasmid construction and recombinant lentivirus packaging

Human JMJD2B was amplified using primers 5′- gcatgcATGGGGTCTGAGGACCACG -3′ (sense) and 5′- atgcatTCATCTGCAAGGGTCTTGAGTTG -3′ (antisense) and cloned into LV5-GFP/Puro vector (Clontech, Mountain View, CA, USA). The clones were confirmed by DNA sequencing. Add the shuttle plasmid and packaging plasmid (LV5-JMJD2B, pRev, pVSV-G) containing the target sequence to 293T cells in proportion. The hJMJD2B shRNA was designed according to the shRNA design rules and other literature (Ye et al., 2012). Bbs I; and BamH I; restriction sites were added to the shRNA end. The designed shRNA sequence was created by Jimma Gene(Shanghai,China).sh-hJMJD2B F: 5′-CACCGCCTGCCTCTAGGTTCATAATTCAAGAGATTATGAACCTAGAGGCAGG TTTTTTG-3′; Sh-hJMJD2B R: 5′-GATCCAAAAAACCTGCCTCTAGGTTCATAATCT CTTGAATTATGAACCTAGAGGCAGGC-3′. hJMJD2B shRNA was amplified and cloned into pGPU6/GFP/Neo vector (Clontech, Mountain View, CA, USA). The clones were confirmed by DNA sequencing. Add the lentiviral vector (Double-stranded GPR78 shRNA) containing the target sequence to 293T cells in proportion.

Transfection

Overexpression and silencing recombinant lentiviruses were purchased from Shanghai Hanheng Biotechnology. Collect hBMSCs cells in logarithmic growth phase, count, resuspend cells in complete medium, adjust the cell concentration to 1 ×105 cells/ml, inoculate into six-well plates, add 2ml of cell suspension to each well, and store at 37 °C, 5% CO2 conditions Incubate overnight. Incubate overnight until the cell density is about 40% to 60%, and carry out virus infection by 1/2 small volume infection method. Aspirate the original medium in the culture well, add 1/2 volume (1 ml) of α-MEM low-glucose medium, and add polybrene to a final concentration of 5 µg/ml, add the corresponding volume of virus stock solution at MOI = 50, shake well, and put Incubate for 4 h at 37 °C, 5% CO2. After 4 h of incubation, add 1/2 volume (1 ml) of α-MEM low-glucose medium to the culture wells, and incubate overnight at 37 °C under 5% CO2 conditions. On the second day after infection, the virus-containing medium was aspirated, replaced with fresh α-MEM low-glucose medium (without polybrene), and the culture was continued at 37 °C, 5% CO2. After hBMSCs were infected with virus for 72 h, the cells were digested and collected, and the cellular protein and RNA were extracted. The expression of JMJD2B in hBMSCs cells was detected by WB and qPCR. The method is the same as before.

Alkaline phosphatase staining

Data were collected as previously described in Wu et al. (2021). Specifically, Alkaline phosphatase staining was performed using a BCIP/NBT Alkaline Phosphatase Color Development Kit (C3206; Beyotime, China) following the manufacturer’s manual. hBMSCs cells were cultured in osteogenic medium for 1, 3, 7 or 14 days, washed with ice-cold PBS, and fixed with 4% paraformaldehyde (Sigma-Aldrich, St. Louis, MO, USA) for 15 min. Add BCIP solution (300X), NBT solution (150X), and BCIP/NBT dyeing working solution in sequence according to the following proportions. Add the BCIP/NBT dyeing working solution obtained after mixing to ensure that the sample can be fully covered. Incubate in the dark at room temperature for 5–30 min or longer (up to 24 h) until the color develops to the expected depth. The ALP positive cells stained blue/purple. For each experiment, a minimum of three washes was counted and the experiments were repeated three times. ALP staining was observed using a 450-fluorescent inverted phase-contrast microscope (Olympus, Tokyo, Japan)

Alizarin red staining

Data were collected as previously described in (Wu et al., 2021). Specifically, cells were cultured in osteogenic medium for 1, 3, 7 or 14 days and then fixed in 95% ice-cold ethanol for 15 min. After three washes with ice-cold PBS, cells were stained with alizarin red (Sigma-Aldrich) for 5 min. Alizarin red staining was observed using an Olympus 450-fluorescent inverted phase-contrast microscope (Olympus, Tokyo, Japan).

Quantitative real-time PCR (qRT-PCR)

Total RNA was isolated from hBMSCs cells using Trizol reagent (DP424, Tiangen, Beijing, China). All quantitative real-time PCR assays were carried out using three technical replicates and three independent cDNA syntheses. Reverse transcription reaction using cDNA synthesis kit (KR118; Tiangen, Beijing , China), qRT-PCR was performed using a fluorescence quantitative PCR kit (QP002, Fulengen, Guangzhou, China) and gene-specific primers (Table 1). GAPDH was used as an internal reference. The relative gene expression was determined using the 2−ΔΔCt method. The PCR reactions were performed in triplicates.

Table 1 Primer sequence details.

Gene name	Gene ID		Primer sequence (5′–3′)	Amplification length (bp)	
h-JMJD2B	NM_001370093.1	Forward:	GGACTAGAGGCCGTCTAAATTG	91	
Reverse:	ACTTCCTGCGTGCAAAGA	
h-Runx2	NM_001015051.3	Forward:	GCTTCATTCGCCTCACAAAC	112	
Reverse:	GTAGTGACCTGCGGAGATTAAC	
h-FAM210A	NM_001098801.2	Forward:	CTGATGGGCGTAAGGAAGAAA	110	
Reverse:	TGGGTCTTTCCCAAGCATAC	
h-GAPDH	NM_001289745.3	Forward:	CAAGAGCACAAGAGGAAGAGAG	102	
Reverse:	CTACATGGCAACTGTGAGGAG	
h-Osteocalcin	NM_199173.6	Forward:	TCACACTCCTCGCCCTATT	114	
Reverse:	CCTCCTGCTTGGACACAAA	
h-Osteopontin	NM_000582.3	Forward:	CATATGATGGCCGAGGTGATAG	108	
Reverse:	AGGTGATGTCCTCGTCTGTA	
h-Osterix	NM_001173467.3	Forward:	CATTCTGGGCTTGGGTATCT	93	
Reverse:	GGCCTGAGATGAGAGTTTGT	
h-Runx2 (ChIP-PCR)	NG_008020	Forward:	TAATCTCCGCAGGTCACTAC	238	
Reverse:	ATACAAACCATACCCAAACC	

Western blot analysis

Data were collected as previously described in (Wu et al., 2021). Specifically, proteins were isolated from hBMSCs cells using RIPA buffer. Protein samples were separated using 5% SDS-PAGE and transferred to a polyvinylidene difluoride (PVDF) membrane. The membrane was incubated with primary antibody against target protein (Table 2), and β-actin (1:2000; B1033; Biodragon) Incubated overnight at 4 °C. The membrane was washed three times with TBST and incubated with HRP-linked secondary antibody (1:1,000; 7074P2; Cell Signaling Technology) for 1 h at room temperature.

Table 2 Antibody details.

	Manu.	Cat.	Isotype	Dilution	
Rabbit anti-KDM4B antiobdy	Abcam	Ab191434	Rabbit	1:1000	
Hrp-Goat Anti-Rabbit IgG	Jackson	111-035-003	Goat	1:10000	
β-actin Polyclonal Antibody	Biodragon	B1033	Rabbit	1:2000	
Rabbit anti-H3K9me2 antibody	Biovision	6814-25	Rabbit	1:1000	
Rabbit Anti-RUNX2 antibody	Abcam	ab76956	Rabbit	1:1000	
Rabbit anti-FAM210A antibody	novusbio	NBP2-13992	Rabbit	1:1000	

Chromatin immunoprecipitation (ChIP)

Treatment of cells with ultrasound cross-linking. ChIP was performed using a ChIP Assay Kit (22202-20; Beyotime, China) following the manufacturer’s instruction. Briefly, cell lysates were centrifuged at 12,000 rpm for 5 min at 4 °C. Take out 20 µl sample as Input for subsequent testing. Add 70 µl protein A+G Agarose/Salmon Sperm DNA to the remaining nearly 2 ml sample (about 35 µl is precipitation and 35 µl is liquid), and mix slowly or shake at 4 °C for 30 min. The supernatant was collected, and incubated with 0.5–1 µg anti-H3K9me2 (6814-25; Biovision) and protein A+G Agarose/Salmon Sperm DNA (of which about 30 µl is precipitation and 30 µl is liquid) overnight at 4 °C. The immunoprecipitates were collected and washed three times with wash buffer. Next, the DNA fragments are purified for subsequent PCR experiments. Wash the sample three times with Elution buffer. Add 20 µl of 5M NaCl to 500 µl of supernatant, mix well, and heat at 65 °C for 4 h to remove crosslinks between protein and genomic DNA. For the 20 µl sample obtained as Input, add 1 µl 5M NaCl, mix well, and heat at 65 °C for 4 h. Then add an appropriate amount of EDTA, proteinase K, Tris, chloroform, glycogen, NaAc, and absolute ethanol for further purification, and the obtained DNA precipitate is used for PCR detection of the target gene.

Immunofluorescence staining

hBMSCS cells were seeded in a 6-well plate at a density of 1 ×105 cells/well. Cells were washed three times with PBS. After treatment, Cells were fixed in 4% paraformaldehyde solution for 15 min at room temperature,followed by fixation with pre-cooled acetone for 15 min. Later, they were washed twice in PBS followed by a 30-minute incubation in 2% bovine serum albuminsolution (Sigma-Aldrich, Cat# A7906-50G). Cells were then treated with 0.5% PBS-Triton X-100 for 20 min and incubated with 3% H2O2 for 15 min. The samples were then blocked with 5% serum for 20 min and incubated with JDJM2B (1:1000; Ab191434; Abcam) and FAM210A(1:200; NBP2-13992; Novusbio) overnight at 4 °C. Then add 200 µL of APC-labeled secondary antibody (1:500) to the sample, and incubate for 1 h at 37 °C, 5% CO2, and saturated humidity. The nucleus was stained with DAPI (S2110; Solarbio) for 1 min. Finally, the samples were washed to remove excess solution and mounted on microscope slides for imaging Images were collected using VistarImage 3.0 software (Nikon, Japan) software connected to a Olympus CKX53 inverted microscope. Images were acquired using an UPlanFLN objective at magnification 40 ×.

Statistical analysis

Data are expressed as the mean ±standard deviation. Statistical analysis was performed using SPSS 23.0 software (SPSS Inc., Chicago, IL, USA). Comparisons among different groups were performed using one-way analysis of variance followed by Tukey’s post hoc test. A P value less than 0.05 was considered statistically significant.

Results

Osteogenesis promotes JMJD2B expression in hBMSCs cells

To investigate the effect of osteogenic medium on hBMSCs proliferation, we treated hBMSCs cells at different time points. To further examine the effect of osteogenic medium treatment on osteogenesis and the involvement of histone modification, we determined the mRNA levels of JMJD2B as well as osteogenesis- and myogenesis- genes in hBMSCs cells at 0, 1, 3, 7, 14 days. The results showed that the osteogenic medium significantly upregulated the mRNA expression of Runx2 during the initial (day 3) osteogenic induction process, and furthermore dramatically upregulated the mRNA expression of FAM210A and JMJD2B at 7 and 14 days simultaneously (Fig. 1A).

Figure 1 The effects of osteogenic medium on osteogenic response of hBMSCs cells.

(A) hBMSCs cells were cultured in osteogenic medium as indicated. Quantitative real-time PCR (qRT-PCR) was performed at d0, d1, d3, d7, and d14 after treatment to measure mRNA expression of JMJD2B as well as osteogenesis- and myogenesis-related genes. GAPDH was used as an internal reference. Data are expressed as the mean ±standard deviation (SD). ∗P < 0.05, ∗∗P < 0.01, ∗∗∗P < 0.001 vs. D0. All experiments were independently repeated at least three times. (B) Western blot analysis was conducted to measure protein expression of Runx2, JMJD2B, FAM210A protein levels. β-actin was used as an internal reference. (C) Immunofluorescence staining was performed based on JMJD2B and FAM210A in hBMSCs. Scale bar = 20 µm.

To further investigate whether osteogenesis affects related protein level. We determined the related protein level in hBMSCs cells treated with osteogenic medium. As shown in Fig. 1A, compared with cells at day 0, the proper osteoinduction time markedly elevated the protein levels of total Runx2, JMJD2B and FAM210A but slightly inhibited Runx2 at day 1, suggesting that osteogenesis may be induced by JMJD2B-mediated histone modification and the expression of FAM210A and Runx2. Next, we detected structure of histone demethylase regulating epigenetic modification in hBMSCs cells to investigated whether JMJD2B-mediated histone demethylation affects hBMSCs differentiation. JMJD2B immunofluorescent staining showed that osteogenesis substantially promoted JMJD2B level at 7, 14 days compared with the control (day 0), whereas no obvious changes in fluorescence signal were shown in 1 day and 3 days. Of note, osteogenesis also significantly promoted FAM210A level (Fig. 1C), which plays a vital role in regulating the structure and function of bones and muscles. These data suggest that osteogenesis may be induced by regulating JMJD2B-mediated histone modification and Runx2 and FAM210A-mediated differentiation of hBMSCs.

Histone H3K9 Demethylase JMJD2B promotes osteogenic differentiation of hBMSCs

To examine whether overexpression of JMJD2B could reverse the effect of silence JMJD2B lentivirus-transfected hBMSCs on osteogenesis, we overexpressed/silenced JMJD2B in hBMSCs cells and then treated the cells with vehicle or osteogenesis Induction Medium. As shown in Figs. 2A and 2B, qRT-PCR and western blot revealed that in general, overexpression of JMJD2B remarkably enhanced mRNA expression and protein level of JMJD2B, while silence of JMJD2B attenuating mRNA/protein expression of JMJD2B (Figs. 2A–2B), regardless of the status of initial protein level. Compared with empty vector transfection (Scr and NC), overexpression of JMJD2B enhanced the intensities of ALP and alizarin red staining, in the presence of osteogenic medium, significantly in 7, 14 days. Silence of JMJD2B treatment resulted in reductions in ALP and alizarin red staining in empty vector-transfected cells. Importantly, slience of JMJD2B reduced ALP and alizarin red staining in the presence of osteogenic medium, suggesting that slience of JMJD2B abrogates the facilitate effects of osteogenic medium on osteogenic differentiation and mineralization of hBMSCs. According to the outcome of staining experiments based on the overexpression of JMJD2B in hBMSCs, that are contrary to silence effect further verify the key role of JMJD2B in the induction of osteogenic differentiation.

Figure 2 JMJD2B regulates the methylation enrichment status of H3K9me2 in RUNX2 promoter to influence the osteogenic differentiation of hBMSCs.

hBMSCs were transfected with empty or JMJD2B-overexpressing/silencing vectors and incubated in osteogenic medium or H3K9me2 inhibitor BIX01294/DMSO for 3, 7 or 14 days hBMSCs were transfected with empty or JMJD2B-overexpressing/silencing vectors and cultured in osteogenic medium (A) hBMSCs were treated for 14 days and the Runx2 promoter was ChIP-ed with anti-H3K9me2 or IgG control. Data are expressed as the mean ±SD. ∗P < 0.05, ∗∗P < 0.01, ∗∗∗P < 0.001, ∗∗∗∗P < 0.0001 vs. LV5-JMJD2B. (B) Alkaline phosphatase (ALP) staining and alizarin red staining were conducted to examine osteogenesis and mineralization, respectively. (C) Quantitative real-time PCR (qRT-PCR) was performed at 3 d after treatment to measure mRNA expression of Runx2 and JMJD2B genes. GAPDH was used as an internal reference. Data are expressed as the mean ±standard deviation (SD). ∗P < 0.05, ∗∗P < 0.01, ∗∗∗P < 0.001. All experiments were independently repeated at least three times. (D–E) Western blot analysis was conducted to measure protein expression of JMJD2B, Runx2, H3K9me2 protein levels. β-actin was used as an internal reference. (F) Quantitative real-time PCR (qRT-PCR) was performed at 3 d after treatment to measure mRNA expression of Runx2 and JMJD2B genes. GAPDH was used as an internal reference. Data are expressed as the mean ±standard deviation (SD). ∗P < 0.05, ∗∗P < 0.01, ∗∗∗P < 0.001. All experiments were independently repeated at least three times.

Then, we measured the mRNA levels of osteogenic markers in hBMSCS cells in response to JMJD2B overexpression/slience. As shown in Fig. 2C, compared with Scr/NC, JMJD2B overexpression dramatically promoted mRNA expression of osteogenic markers, including Runx2, Ocn, Opn, and Osx, whereas JMJD2B slience generally showed counter effects on mRNA expression of these genes (Fig. 1C, upper and middle panels), regardless of the presence of osteogenic medium. Collectively, the combined results of Figs. 1 and 2 suggest that histone H3K9 Demethylase JMJD2B might promotes osteogenic differentiation of hBMSCs by regulating the expression of Runx2,Ocn,Opn,Osx.

JMJD2B regulates the methylation enrichment status of H3K9me2 in RUNX2 promoter to influence the osteogenic differentiation of hBMSCs

Considering the critical role of JMJD2B in osteogenesis and the essential role of histone modification in hBMSCs differentiation regulation, we examined the effect of JMJD2B overexpression/slience on the methylation enrichment status of H3K9me2 in the Runx2 promoter in hBMSCs exposed to osteogenic medium, we wanted to determine whether the altered transcriptional activity of JMJD2B may be attributed to differences in DNA binding. To do this we compared the chromatin association of H3K9me2 with the Runx2 promoter using Chromatin Immunoprecipitation (ChIP). LV5-JMJD2B significantly increased H3K9me2 occupancy at the Runx2 promoter. However, Si-JMJD2B decreased H3K9me2 recruitment to Runx2 promoter (Fig. 3A). These results strongly suggest that LV5-JMJD2B has a different activity of co-regulator recruitment than Si-JMJD2B. The ChIP results revealed that binding levels of H3K9me2 at the promoter regions of Runx2 were significantly increased when JMJD2B overexpressed, whereas they were reduced at the promoter region while slienced (Fig. 3A), regardless of the presence of osteogenic medium.

Figure 3 JMJD2B Is Required for Osteogenic Differentiation of hBMSCs.

hBMSCs were transfected with empty or JMJD2B-overexpressing/silencing vectors and incubated in osteogenic medium for 0, 1, 3, 7 or 14 days (A) Quantitative real-time PCR (qRT-PCR) was performed at 72 h after treatment to measure mRNA expression of JMJD2B as well as osteogenesis-related genes. GAPDH was used as an internal reference. Data are expressed as the mean ±standard deviation (SD). ∗P < 0.05, ∗∗P < 0.01, ∗∗∗P < 0.001 vs. 0 M. All experiments were independently repeated at least three times. ALP staining and alizarin red staining were performed to examine osteogenesis and mineralization, respectively (B) Western blot analysis was conducted to measure protein expression of JMJD2B protein levels. β-actin was used as an internal reference. (C) Alkaline phosphatase (ALP) staining and alizarin red staining were conducted to examine osteogenesis and mineralization, respectively. (D) Quantitative real-time PCR (qRT-PCR) was performed at 3 d after treatment to measure mRNA expression of osteogenesis- related genes. GAPDH was used as an internal reference. Data are expressed as the mean ±standard deviation (SD). ∗P < 0.05, ∗∗P < 0.01, ∗∗∗P < 0.001. All experiments were independently repeated at least three times.

As shown in Fig. 3B, compared with control, H3K9me2 inhibitor BIX01294 impaired the intensities of ALP and alizarin red staining, in the presence of osteogenic medium. Osteogenic medium treatment resulted in enhancement in ALP and alizarin red staining in control cells. Importantly, inhibition of H3K9me2 reduced ALP and alizarin red staining in the presence of osteogenic medium, suggesting that inhibition of H3K9me2 abrogates the promotion effects of osteogenic medium on osteogenic differentiation and mineralization of hBMSCs.

To further investigate whether JMJD2B activates H3K9me2 binding levels, we measured the mRNA levels of Runx2 and JMJD2B in hBMSCs cells in response to H3K9me2 inhibition. As shown in Fig. 3C, compared with control, H3K9me2 inhibition dramatically reduced mRNA expression of Runx2 and JMJD2B. Next, we determined the protein level in hBMSCs cells inhibiting H3K9me2. As shown in Fig. 3D, compared with control, inhibition of H3K9me2 markedly cut down the protein levels of Runx2 and H3K9me2 but showed no significantly effects on JMJD2B in the presence of osteogenic medium. To examine whether compensatory expression of JMJD2B could reverse the effect of BIX01294 on these proteins, we overexpressed JMJD2B in hBMSCs and then treated the cells with DMSO medium dilution solution or BIX01294. Treatment with LV5-JMJD2B significantly overexpressed JMJD2B and elevated protein levels of Runx2 and JMJD2B in the absence or presence of BIX01294. Of note, inhibition of H3K9me2 significantly reduced Runx2 and H3K9me2 protein levels but does not affect the compensatory increase of JMJD2B overexpressing JMJD2B (Fig. 3E). qRT-PCR revealed that in general, overexpression of JMJD2B remarkably enhanced mRNA expression of Runx2 and JMJD2B.This enhancement slightly hindered by BIX01294 (Fig. 3F). These results suggest that by affecting the methylation enrichment status of H3K9me2 in the Runx2 promoter, JMJD2B may regulate the expression of Runx2 and mediate the differentiation of hBMSCs.

Discussion

DNA is the essence of heredity. This view has almost become a genetic truth. The rise of epigenetics has changed this phenomenon to a certain extent. People are surprised to find that the emergence of new traits is not necessarily the result of changes in DNA sequences. The environment may induce changes in biological traits, and this change can also Inherited to the next generation. Epigenetic refers to genetic changes in gene expression without altering the DNA sequence, including DNA methylation, histone modification and small non-coding RNA related regulation (Wu & Sun, 2006). Lysine methylation is one of the most prominent histone post-translational modifications that regulate chromatin structure (Berry & Janknecht, 2013). All four members of the JMJD2 family (also known as KDM4) have the ability to demethylate H3K9me2/3 and/or H3K36me2/3. The JMJD2 family belongs to a group of proteins containing the Jmj domain. JMJD2 family members are involved in the regulation of stem cell self-renewal and differentiation (Mak et al., 2021). Of note, JMJD2B also participates in the osteogenic differentiation of mesenchymal stem cells by mediating the demethylation of H3K9me3 on the DLX promoter (Ye et al., 2012). Chromatin immunoprecipitation (ChIP) coupled with PCR (ChIP-PCR) also identified a STAT-binding site in Runx2 promoter region containing a transcriptional repressor trimethylated histone 3 lysine 9 (H3K9me3), which can be demethylated by JMJD2B (Kurozumi et al., 2019).These findings suggest that histone H3K9 Demethylase JMJD2B might promotes osteogenic differentiation of bone marrow-derived mesenchymal stem cells by regulating H3K9me2 on Runx2, JMJD2B possibly playing an important role in bone formation and resorption.

To verify the effect of JMJD2B on expression and osteogenic response, we treated hBMSCs with osteogenic medium. We found that mRNA/protein/fluorescence signal expression of JMJD2B was significantly upregulated in response to osteogenic medium stimulation, along with significantly facilitated/suppressed osteogenic differentiation in the following gain/loss of function study, suggesting that osteogenic medium promotes osteogenesis possibly by upregulating JMJD2B-mediated histone demethylation of osteogenesis-related chromatin. The canonical transcription factor Runx2 facilitates bone formation. Of note, JMJD2B-mediated histone demethylation is essential for Runx2 expression. Thus, we hypothesized that osteogenic medium promotes osteogenesis possibly through upregulating JMJD2B-mediated histone demethylation of H3K9me3/H3K9me2 in Runx2 promoter and that inhibitory expression of H3K9me2 might suppress the effects of osteogenic medium. Indeed, our results showed that overexpression/silence of JMJD2B increased/reduced the enrichment status of H3K9me2 in Runx2 promoter, in addition, inhibitory expression of H3K9me2 reversed the promoted effects of osteogenic medium on osteogenic differentiation as well as mRNA/protein expression of Runx2,which could be compensated by overexpression of JMJD2B. Together, these events facilitate osteogenic differentiation of hBMSCs, as evidenced by enhanced ALP and alizarin red staining and mRNA expression of osteogenic markers in hBMSCs. JMJD2B-mediated histone demethylation plays an essential role in the activation of Runx2 expression. Dysregulation of JMJD2B blocks the activation of Runx2 expression.

Conclusions

In conclusion, osteoinduction promotes JMJD2B expression in hBMSCs. Overexpression of JMJD2B reversed the inhibitory effects of negative inhibition of silent lentivirus on osteogenic differentiation as well as mRNA expression of osteogenic markers in hBMSCs. JMJD2B regulates the methylation enrichment status of H3K9me2 in the Runx2 promoter to influence the osteogenic differentiation and Runx2 expression in hBMSCs. These findings suggest that overexpression of JMJD2B is as a potential therapeutic approach to regulate the osteogenic differentiation of hBMSCs through activating histone demethylation of H3K9me3/H3K9me2 in the Runx2 promoter.

Supplemental Information

Supplemental Information 1 Raw data applied for data analyses and preparation for Fig. 1B for the time period of 0, 1, 3, 7, 14D

Click here for additional data file.

Supplemental Information 2 Raw data applied for data analyses and preparation for Fig. 2B

Click here for additional data file.

Supplemental Information 3 Raw data applied for data analyses and preparation for Fig. 3D

Click here for additional data file.

Supplemental Information 4 Raw data applied for data analyses and preparation for Fig. 3E

Click here for additional data file.

Supplemental Information 5 Raw data exported from the microscope CKX41 applied for data analyses and preparation for Fig. 1C FAM210a for the time period of 0, 1, 3, 7, 14D

Click here for additional data file.

Supplemental Information 6 Raw data exported from the microscope CKX41 applied for data analyses and preparation for Fig. 1C JMJD2B for the time period of 0, 1, 3, 7, 14D

Click here for additional data file.

Supplemental Information 7 Histograms and raw data showing mRNA fold expression and grayscale value of RUNX2, JMJD2B, FAM210A for Fig. 1

Click here for additional data file.

Supplemental Information 8 Histograms and raw data showing mRNA fold expression and grayscale value of JMJD2B and other osteogenesis-related genes for the time period of 72 h for Fig. 2

Click here for additional data file.

Supplemental Information 9 Raw data exported from the microscope applied for data analyses and preparation for Fig. 2C left panel Alkaline phosphatase (ALP) staining for the time period of 1, 3, 7, 14D

Click here for additional data file.

Supplemental Information 10 Raw data exported from the microscope applied for data analyses and preparation for Fig. 2C left panel Alizarin red staining for the time period of 1, 3, 7, 14D

Click here for additional data file.

Supplemental Information 11 Raw data exported from the microscope applied for data analyses and preparation for Fig. 3B left panel Alkaline phosphatase (ALP) staining for the time period of 7D

Click here for additional data file.

Supplemental Information 12 Raw data exported from the microscope applied for data analyses and preparation for Fig. 3B left panel Alizarin red staining for the time period of 14D

Click here for additional data file.

Supplemental Information 13 Histograms and raw data showing CHIP-qPCR, mRNA fold expression and grayscale value of Fig. 3

Click here for additional data file.

Supplemental Information 14 Statistical Reporting

Click here for additional data file.

We thank Meilan Chen for her encouragement.

Additional Information and Declarations

Competing Interests

Author Contributions

Data Availability

The authors declare there are no competing interests.

Pan Kang analyzed the data, prepared figures and/or tables, and approved the final draft.

Zhiming Wu analyzed the data, prepared figures and/or tables, and approved the final draft.

Yuxi Huang analyzed the data, prepared figures and/or tables, and approved the final draft.

Zhen Luo performed the experiments, authored or reviewed drafts of the article, and approved the final draft.

Shaochuan Huo performed the experiments, analyzed the data, authored or reviewed drafts of the article, and approved the final draft.

Qunqun Chen conceived and designed the experiments, authored or reviewed drafts of the article, and approved the final draft.

The following information was supplied regarding data availability:

Raw data are available in the Supplementary Files.

All data presented in histograms including qRT-PCR, Western blot and CHIP-qPCR, were plotted and counted by Graphpad Prism 8.0 and are available in the PZFX file format. The original software can be found at https://www.graphpad.com/.

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
