# Peer review of "Histone H3K9 demethylase JMJD2B/KDM4B promotes osteogenic differentiation of bone marrow-derived mesenchymal stem cells by regulating H3K9me2 on RUNX2"

_PeerJ, doi:10.7717/peerj.13862_

## Round 0.1 · original submission · Major Revisions

Please address all the issues raised by the reviewers.

·

Basic reporting

-The aim of the study was not written clearly, only a wide results section was started to be explained. Abstract and the purpose should be added to the necessary places with a sentence. It is very important for further reading and benefiting of this study. However, at first glance, the article does not reflect the scope and purpose.
-English and spelling mistakes of the study should be reviewed again.
-In the material method part of the study, the explanation of some studies was insufficient.
-The resolution of graphics and figures is not good.

Experimental design

It is well planned as an experimental design.

-Primers of h-JMJD2B, h-Runx2, h-GAPDH genes were repeated in the primer list in Table 1.Is there any particular reason for this? If any, it should be disclosed, otherwise it should be removed.

-It is not clear which primer was used for the ChIP technique.

Validity of the findings

Some deficiencies in the findings section prevent good interpretation of the results.

-Figure 1, figure2 and figure 3 resolutions and visuals are very bad, they need to be rearranged and added. This state is not suitable for this study.

-There is no results for sample obtained as Input DNA in figure3A.

-The quality of the fluorescent pictures is very poor and the background colors are not the same.

-Alkaline phosphatase (ALP) staining and Alizarin red staining images in Figure 2C are not clear.

Reviewer 2 ·

Basic reporting

This is an interesting study where authors aimed to demonstrate if JMJD2B/KDM4B induces osteogenic differentiation of hBMSCs by regulating the methylation level of H3K9me2 at the Runx2 promoter. The main strength of this study is the use of different methods such as alizarin red and alkaline phosphatase staining to detect osteogenic differentiation and mineralization ability, as well as the use of western blot and quantitative RT-PCR to investigate osteogenesis-related proteins and expression levels of different genes respectively and, other complementary methods. However, in my opinion, it requires a major revision before it can be considered for publication in PeerJ.
- There are also numerous typo errors and grammatical issues throughout the manuscript.

- Abbreviations should be defined where appear first in the manuscript.

- The same abbreviation is used in different ways in different parts of the manuscript. For example, "human bone marrow mesenchymal stem cells” is written as (hBMSCs) in line 60 and as (HBMSCs) in line 65.

Experimental design

The major concerns related to the current experiment are as follow;
- The purpose of the study is not explained clearly neither in the abstract nor in the introduction part. The gap in the literature and the purpose of the study are not explained clearly. It seems that the current literature in this field has not been evaluated well. It is not mentioned clearly which shortcomings of the previous studies in this field will be enlightened and eliminated with the data obtained from this research. The expression in the last sentence of the introduction is insufficient to express the purpose of the researchers (line112-113).

- GAPDH was used as a reference gene in qRT-PCR experiments. However, this gene is not considered to be the more stable gene in bone marrow mesenchymal stem cells and the osteogenic differentiation process. It is recommended that the authors validate their work with another reference gene. (https://doi.org/10.1155/2019/3093545)

- It is stated that mRNA expression of JMJD2B and osteogenesis-related genes were investigated 72 hours after treatment. The reason why 72 hours was chosen for the experiments is not explained. If the expression of these genes had been investigated at the osteogenesis stage in samples from days 0, 1, 3, 7, or 14, more meaningful and reliable data could have been achieved in line with previous experiments.

Validity of the findings

no comment

Annotated reviews are not available for download in order to protect the identity of reviewers who chose to remain anonymous.

---

## Round 0.2 · Minor Revisions

The issue raised by the reviewer on the reference gene should be addressed.

·

Basic reporting

The article meets the corrections and suggestions given and should be accepted as is.

Experimental design

Experimental design meets recommendations

Validity of the findings

The findings are appropriate and sufficient.

Additional comments

The article meets the corrections and suggestions given and should be accepted as is.

Reviewer 2 ·

Basic reporting

The manuscript is clear and unambiguous, but there are also numerous typo errors and grammatical issues throughout the manuscript.

Literature references and adequate space infrastructure have been established.

Professional article structure, figures, tables. Raw data are presented.

Experimental design

The research question is well defined, relevant and meaningful. It is stated how research fills an identified knowledge gap.
Methods were described with sufficient detail, but still, the following comment has not been explained by the authors or the necessary changes have not been made.
GAPDH was used as a reference gene in qRT-PCR experiments. However, this gene is not
considered to be the more stable gene in bone marrow mesenchymal stem cells and the
osteogenic differentiation process. It is recommended that the authors validate their work with
another reference gene. (https://doi.org/10.1155/2019/3093545)

Validity of the findings

All underlying data have been provided; they are robust, statistically sound, and controlled.

Conclusions are well stated, and linked to the original research question.

---

## Round 0.3 · accepted · Accept

The answers to the reviewers are acceptable.